# Reconstruction of High-Grade Trochlea Dysplasia in a Young Female with Recurrent Patella Dislocation: A Case Report

**DOI:** 10.3390/medicina59050986

**Published:** 2023-05-19

**Authors:** Chih-Hsuan Wu, Kuo-Yao Hsu, You-Hung Cheng, Cheng-Pang Yang, Huan Sheu, Shih-Sheng Chang, Chao-Yu Chen, Chih-Hao Chiu

**Affiliations:** 1Department of Orthopedic Surgery, Linkou Chang Gung Memorial Hospital, Taoyuan 333, Taiwan; shinywu220@gmail.com (C.-H.W.);; 2Department of Orthopedic Surgery, New Taipei Municipal Tucheng Hospital, New Taipei City 236, Taiwan; 3Department of Orthopedic Surgery, Taoyuan Chang Gung Memorial Hospital, Taoyuan 333, Taiwan

**Keywords:** patella dislocation, trochlea dysplasia, Dejour classification

## Abstract

The patellofemoral joint involves a combination of bony structures and soft tissues to maintain stability. Patella instability is a disabling condition, and the cause is multifactorial. The main risk factors include patella alta, trochlea dysplasia, excessive tibial tuberosity to trochlea grove (TT–TG) distance, and excessive lateral patella tilt. In this case report, we highlight the thinking process of diagnosis and method for selecting the optimal treatment in accordance with the guidelines by Dejour et al. when we are presented with a patient with patella instability. A 20-year-old Asian woman without underlying medical conditions, presented with recurrent (>3 episodes) right patella dislocation for 7 years. Investigations revealed a type D trochlea dysplasia, increased TT–TG distance, and excessive lateral tilt angle. She underwent trochlea sulcus deepening, sulcus lateralization and lateral facet elevation, lateral retinacular release, and medial quadriceps tendon–femoral ligament (MQTFL) reconstruction. Due to the complexity behind the anatomy and biomechanics of patella instability, an easy-to-follow treatment algorithm is essential for the treating surgeon to provide effective and efficient treatment. MQTFL reconstruction is recommended for recurrent patella dislocation due to satisfactory clinical and patient reported outcomes and a reduced risk of iatrogenic patella fracture. Controversies for surgical indication in lateral retinacular release, and whether the sulcus angle is an accurate parameter for diagnosis of trochlea dysplasia, remain, and further research is required.

## 1. Introduction

Patella instability is a disabling condition commonly affecting the female population in their second and third decades of life [1,2]. The patellofemoral joint relies on a combination of bony structures and soft tissues to maintain stability [1,2,3]. The geometry of the patella and trochlea has been shown to affect patella tracking and stability [2]. Soft tissue attachments to the patella contributes to the active stabilizing forces through the vastus medialis obliquus (VMO) and passive stabilizers, principally through the medial patellofemoral ligament (MPFL) and attachments of the lateral retinaculum [1,2].

The dislocation rate of the patella for the first time is 17–20% and as high as 44–70% with consecutive dislocations [2]. The cause for patella instability is multifactorial [1]. Previous patella dislocation is believed to be the greatest risk factor for patella instability. Of the multiple risk factors, Dejour et al. proposed the four major culprits for patella instability were patella alta, trochlea dysplasia, excessive tibial tuberosity to trochlea grove (TT–TG) distance, and excessive lateral patella tilt [4].

Throughout the years, the literature has continuously been presented with innovative advances and controversies for the diagnosis and management of patella instability. In 2021, Dejour et al. published a treatment guideline that highlights and proposes treatment recommendations for each of the four main risk factors [5]. Due to the complexity behind patella instability, this easy-to-follow treatment algorithm allows the treating surgeon to manage the patient effectively without missing essential components.

In this case report, we highlight the thinking process of diagnosis and method for selecting the optimal treatment in accordance with the guidelines when we are presented with a patient with patella instability.

## 2. Clinical Cases

### 2.1. Case

A 20-year-old Asian woman without underlying medical conditions, presented with recurrent (>3 episodes) right patella dislocation for 7 years. Physical examination revealed patellofemoral arthrosis, positive apprehensive test, positive J-sign [5] and high-grade patella instability [4,5]. Anteroposterior and lateral radiographs of the knee (Figure 1), as well as a standing scintinography (Figure 2), did not reveal varus or valgus malalignment (defined by hip knee angle (HKA) [6] of 1.3; quadriceps angle (Q angle) [7] of 10°). Sulcus angle [4] of right and left knees were 138° and 132°, respectively, with lateral subluxation of the right patella (Figure 3). There was no evidence of patella alta (Caton Deschamps ratio 1.03 [8], Figure 4a; Insall-Salvati ratio 1.07 [9], Figure 4b).

Computed tomography (CT) and magnetic resonance imaging (MRI) confirmed suspicion of trochlea dysplasia. Patellotrochlear index (PTI) [10] on MRI was 0.36 (Figure 5). Lateral patella tilt [4] measured on CT was 28.7° (Figure 6a). The TT–TG distance measured on MRI was 22.07 mm (Figure 6b) [2,11]. There was a presence of a supratrochlear spur, hypoplastic medial facet and convex lateral facet with a vertical ‘cliff’ between the two facets (Figure 7). Under the impression of Dejour classification type D trochlea dysplasia [1], a surgical operation was arranged.

### 2.2. Surgical Procedure

She underwent trochlea sulcus deepening, sulcus lateralization and lateral facet elevation, lateral retinacular release and medial quadriceps tendon–femoral ligament (MQTFL) reconstruction. Postoperative recovery was smooth and without complication. Figure 8 and Figure 9 highlight the patient’s postoperative recovery on day 1 and at 2 months which demonstrate a satisfactory outcome. The follow-up period was 6 months to-date.

## 3. Discussion

Dejour et al. published an updated treatment guideline (Figure 10) for patella instability in 2021 [5]. This guideline is an algorithm that highlights each of Dejour’s four major risk factors of patella instability to be evaluated such as “a la carte” items on a menu and the recommended management options recommended for each [5].

With consideration to the above guidelines, surgical treatment is recommended for our patient because she had more than three dislocations.

### 3.1. Trochlea Dysplasia

Characteristic signs of trochlea dysplasia are best seen on a lateral standing plain radiograph [4]. The crossing sign represents the exact location where the line of the anterior femoral condyles crosses the trochlea groove meaning that the trochlea becomes flat in this exact location [4]. A double contour sign represents a hypoplastic medial facet [4]. The supratrochlear spur is a global prominence of the trochlea and arises in the proximal aspect [4]. It corresponds to a compensatory attempt to contain the lateral displacement of the unstable patella during knee motion [4].

The Dejour Classification classifies trochlea dysplasia into four types based on the three characteristic signs [1,2] (Figure 11). Type A dysplasia has a crossing sign and a trochlea that is shallower than normal, but its morphology remains symmetrical and concave [1,2]. Type B dysplasia has a crossing sign, supratrochlear spur and a flat trochlea [1,2]. Type C has both crossing and double contour signs [1,2]. There is no supratrochlear spur [1,2]. The lateral facet of the trochlea is convex-shaped and the medial facet is hypoplastic [1,2]. Type D has a crossing sign, double contour sign and supratrochlear spur [1,2]. In axial view of the trochlea, there is the presence of a vertical ‘cliff sign’ demonstrating clear asymmetry of the height of the lateral and medial facets [1].

Our patient had type D trochlea dysplasia as evidenced on imaging with the presence of a crossing and double contour sign, supratrochlear spur and a vertical ‘cliff’ between a hypoplastic medial facet and convex lateral facet. Sulcus deepening trochleoplasty is recommended for type D trochlea dysplasia [5]. In our patient, we performed a combination of trochlea sulcus deepening, sulcus lateralization and lateral facet elevation to maximise the trochlea sulcus groove.

### 3.2. Patella Alta

Patella alta is a patella that rides abnormally high in relation to the trochlea groove, with reduced patellofemoral articular contact, and increases the extent of knee flexion before the patella can engage the trochlea groove [1,2,8,12,13].

Patella height can be determined by various methods, though, to date, there is no consensus on the most reliable measurement method [8]. The Caton–Deschamps method is measured by dividing the distance from the inferior articular surface of the patella to the anterosuperior margin of the tibia plateau (PTG) by the length of the patella articular surface (PG) [8]. Patella alta is defined as a ratio greater than 1.3 [8,9,14]. Insall–Salvati method measures the ratio of length of patella tendon (LT) to length of patella (LP). A ratio more than 1.2 suggests patella alta [9].

The cartilage of the patellofemoral joint is evident on a sagittal MRI, which allows patella height to be assessed using the patella–trochlea cartilage overlap [8,15]. This is a critical clinical factor because there is a reduced area of patellofemoral articular contact with patella alta [8,15]. Patellotrochlear index (PTI) is a parameter used for determining the degree of patella–trochlea cartilage overlap on MRI, and is measured by the ratio of the length of trochlea cartilage overlapping the patella cartilage to the length of the patella cartilage [10]. Normal range is between 0.18 and 0.80 [16]. A ratio less than 0.18 is suggestive of patella alta [16].

Using the above methods and parameters for calculation, our patient does not have patella alta (Caton–Deschamps method (PTG:PG) = 33.62:32.49 = 1.03; Insall–Salvati method (LT:LP) = 42:38.95 = 1.07; PTI value was 0.36), and therefore tibia tubercle distalisation was not performed.

### 3.3. TT–TG Distance

TT–TG distance is used to determine the degree of lateralisation of the tibia tubercle in relation to the deepest part of the trochlea groove, that is, the alignment of the extensor mechanism [2,11]. It is measured as the distance between two perpendicular lines from the posterior intercondyler line to the tibia tubercle (TT) and trochlea groove (TG) [2,11]. A distance greater than 20 mm is considered abnormal on CT and greater than 13 mm on MRI [4,5,17,18].

There is no consensus on which image modality (CT versus MRI) is considered the gold standard to measure this parameter. Whilst excellent interobserver reliability is demonstrated on both CT and MRI (ICC 0.777 vs. 0.843, respectively), the values derived from these two tests may not be interchangeable (ICC 0.532 and 0.539, respectively) [19]. Patient positioning and techniques amongst different imaging centres may also influence the measurement [19,20,21].

For our study, TT–TG distance was measured on MRI imaging. Whilst a TT–TG distance of 22.07 mm in our patient suggests that tibial tubercle medialization should be performed, it was not enacted in our case because we believed the main contributor to the increased TT–TG distance was a proximal malalignment; therefore, both lateralization, in addition to deepening of the new sulcus, was performed to correct proximal alignment and hence TT–TG distance.

### 3.4. Lateral Patella Tilt

Lateral patella tilt is confirmed if the angle between the posterior condylar axis and mid-patella line is greater than 20° and suggests trochlea dysplasia [2,4]. In Dejour’s treatment guidelines, the current literature highlights that this parameter has lost its previous role and is not used anymore [1,5,22]. Lateral release is rarely performed in isolation due to its inability to align the patella more medially [1,22]. Although there is controversy regarding surgical indications, lateral release is recommended in patients who have clinical lateral tightness, otherwise defined as negative medial patella tilt [5,23].

In our patient, lateral patella tilt measured on CT was 28.7° and intraoperatively, excessive lateral tightness was examined so lateral release was performed in conjunction with the other procedures.

### 3.5. MPFL and MQTFL Reconstruction

A systematic review demonstrated that MPFL reconstruction is recommended particularly for patients with recurrent dislocations, as the MPFL can contribute up to 60% of resistance to lateral displacement of the patella [5,24]. The goal of MPFL reconstruction is to restore normal patellofemoral kinematics [24]. A case study of isolated MPFL repair for recurrent instability found 8 of 29 knees (28%) at a minimum of 2 years follow up had a later recurrence [25]. MPFL reconstruction is associated with a complication rate of 16.2%, including recurrent lateral patella instability, knee motion stiffness with flexion deficits, patella fractures (3.4% of cases) and patellofemoral arthrosis and pain. Of these complications, 47% were secondary to preventable technical factors [26].

In contrast to MPFL reconstruction, no transosseous patella drilling or bone tunnels are required for suture anchor placement for MQTFL reconstruction. Transosseous drilling and bone tunnels in the patella increase the risk for development of iatrogenic patella fractures. In Spang et al.’s comparative study of the patellofemoral kinematics and stability between four conditions: an intact MPFL, transected MPFL, reconstructed MPFL and reconstructed MQTFL [27], they found that MQTFL restored stability to lateral translation, whereas MPFL had significantly less lateral translation than other states [27]. Peak contact pressures were significantly lower for both reconstructed ligaments [27]. At full extension, both reconstructed MPFL and MQTFL showed significantly lower external patella rotation compared with the transected state, though there were no significant differences at higher flexion angles [27]. With regard to kinematics, the transected state led to a more lateral patella position at full extension, whereas reconstructed MPFL and MQTFL led to more medial positions [27].

We performed a MQTFL reconstruction in our patient due to satisfactory clinical and patient reported outcomes in the literature, and the reduced risk of iatrogenic patella fracture.

### 3.6. Sulcus Angle

The sulcus angle measured on merchant view of the knee is a commonly used parameter for the diagnosis of trochlea dysplasia [2,4,28]. It is defined as the angle between the medial and lateral facets of the trochlea groove, and an angle greater than 145° is deemed abnormal [2,4,28].

However, in our case study we have noted that the sulcus angle on a knee merchant view may not be a reliable criterion for Dejour types B–D trochlea dysplasia as widely reported in the literature. This is because the knee is flexed at 45° on a merchant view, well below the position of the trochlea. This could explain why our patient who presented with Dejour type D trochlea dysplasia had a sulcus angle of 138°, despite the literature pointing out abnormalities exist when the angle is greater than 145°.

### 3.7. Limitations

The main limitation of this case report is the inclusion of one single case that is typical for patellofemoral instability. However, a typical case allows readers to generalise the results when they are presented with patients of similar clinical conditions. A follow-up period longer than 6 months would be more satisfactory.

## 4. Conclusions

Due to the complexity behind the anatomy and biomechanics of patella instability, an easy-to-follow treatment algorithm is essential for the treating surgeon to provide effective and efficient treatment without fear of missing essential components. MQTFL, as an alternative to MPFL reconstruction, is recommended for recurrent patella dislocation due to satisfactory clinical and patient reported outcomes and a reduced risk of iatrogenic patella fracture. Controversies for surgical indication in lateral retinacular release, and whether the sulcus angle is an accurate parameter for diagnosis of trochlea dysplasia, remain, and further research is required.

## Figures and Tables

**Figure 1 medicina-59-00986-f001:**
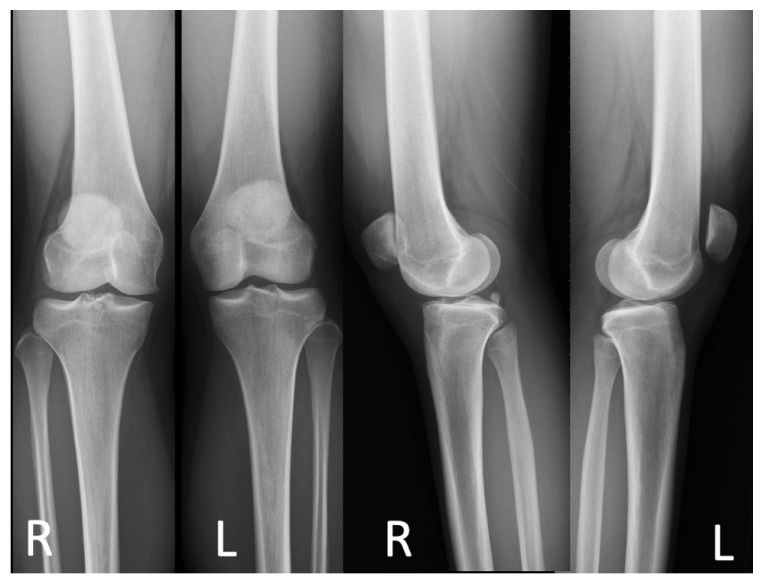
Preoperative standing anteroposterior and lateral plain radiographs of the patient’s knees. R, right; L, left.

**Figure 2 medicina-59-00986-f002:**
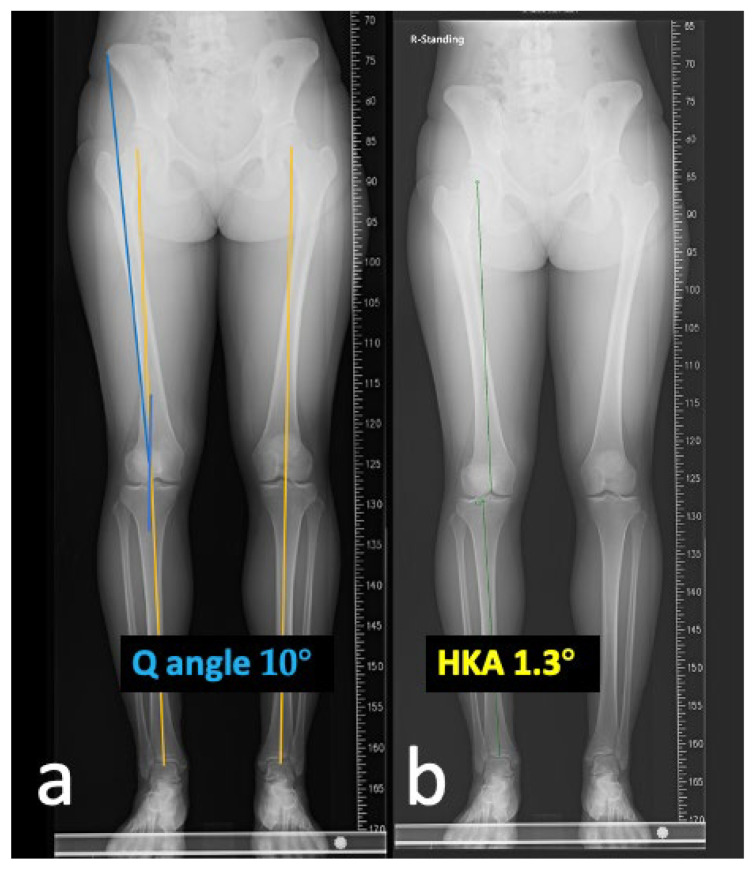
Standing scintigraphy reveals a (**a**) Q-angle of 10° without obvious valgus or varus malalignment; (**b**) Hip knee angle (HKA) of 1.3°.

**Figure 3 medicina-59-00986-f003:**
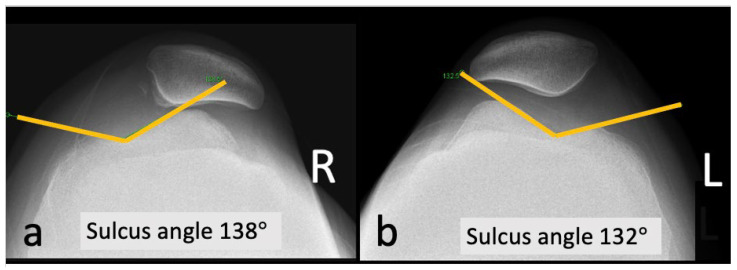
Sulcus angle of (**a**) right knee was 138° and (**b**) left knee was 132°; R, right; L, left.

**Figure 4 medicina-59-00986-f004:**
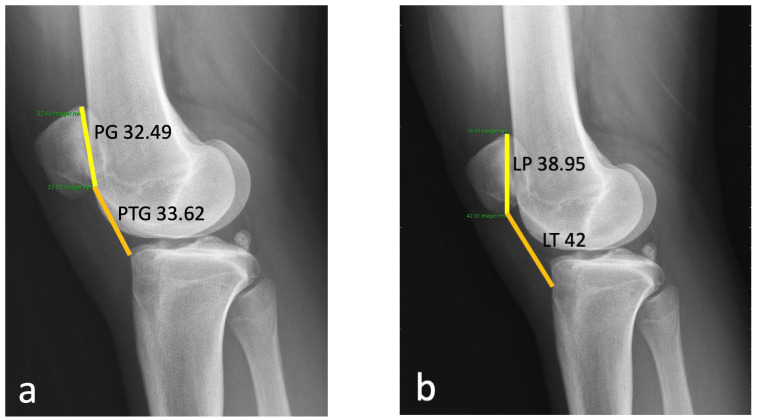
(**a**) Measurement of patient’s patella height using Caton–Deschamps method (PTG:PG) = 33.62:32.49 = 1.03; PTG: line between anterior angle of tibia plateau to the most inferior aspect of patella articular surface, PG = patella articular cartilage surface. (**b**) Measurement using Insall–Salvati method (LT:LP) = 42:38.95 = 1.07; LT = length of patella tendon, LP = length of patella.

**Figure 5 medicina-59-00986-f005:**
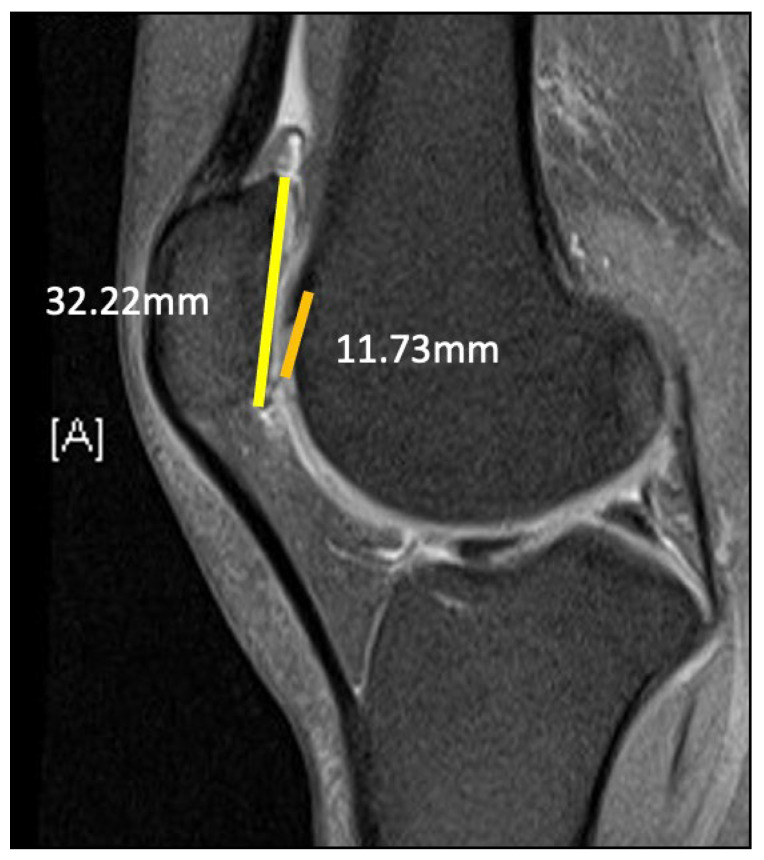
Right knee sagittal MRI. Articular overlap (PTI, patellotrochlear index) was measured by the length of trochlea cartilage overlapping patella cartilage divided by overall patella cartilage length; in our patient, the PTI value was 0.36.

**Figure 6 medicina-59-00986-f006:**
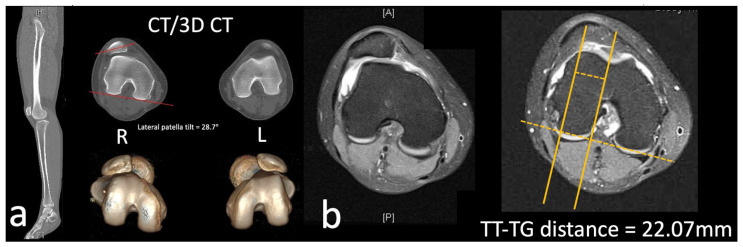
(**a**) Determination of lateral patella tilt on CT; lateral patella tilt in our patient was 28.7°. (**b**) Measurement of TT–TG distance on MRI; in our patient, TT–TG distance was 22.07 mm. CT, computed tomography; TT–TG, tibial tubercle to trochlea groove; R, right; L, left.

**Figure 7 medicina-59-00986-f007:**
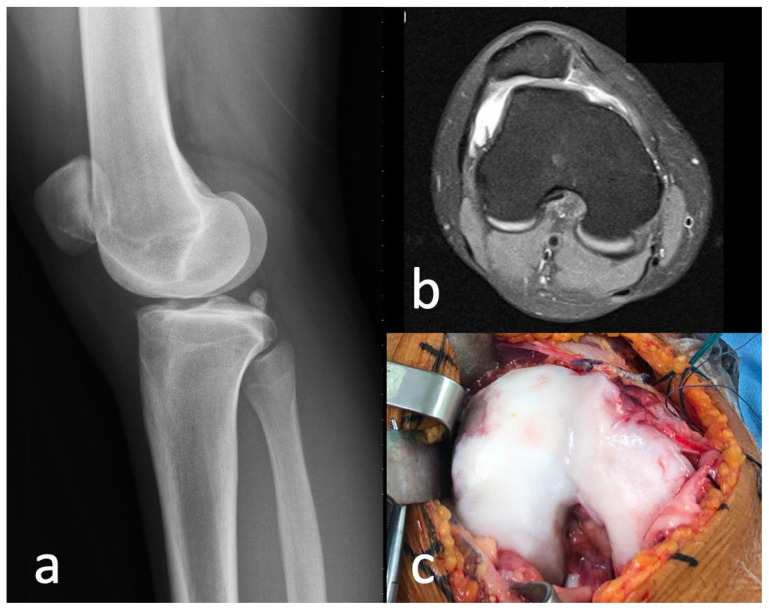
(**a**) Lateral radiograph of right knee revealing crossing and double contour signs, and a supratrochlear spur. (**b**) MRI image of trochlea dysplasia. (**c**) Intraoperative image of trochlea revealing “vertical cliff” link between a convex lateral facet and hypoplastic medial facet.

**Figure 8 medicina-59-00986-f008:**
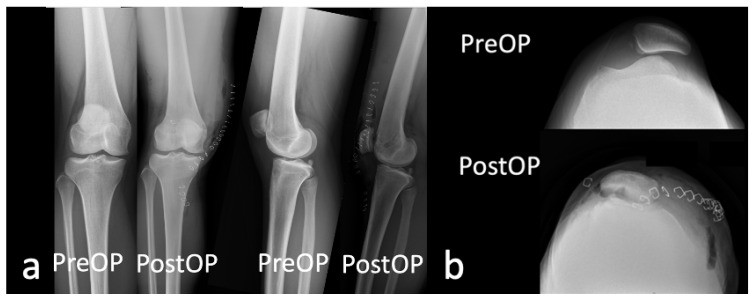
Comparison of the patient’s preoperative and postoperative (**a**) anteroposterior, lateral and (**b**) merchant knee plain radiographs at post-operative day 1.

**Figure 9 medicina-59-00986-f009:**
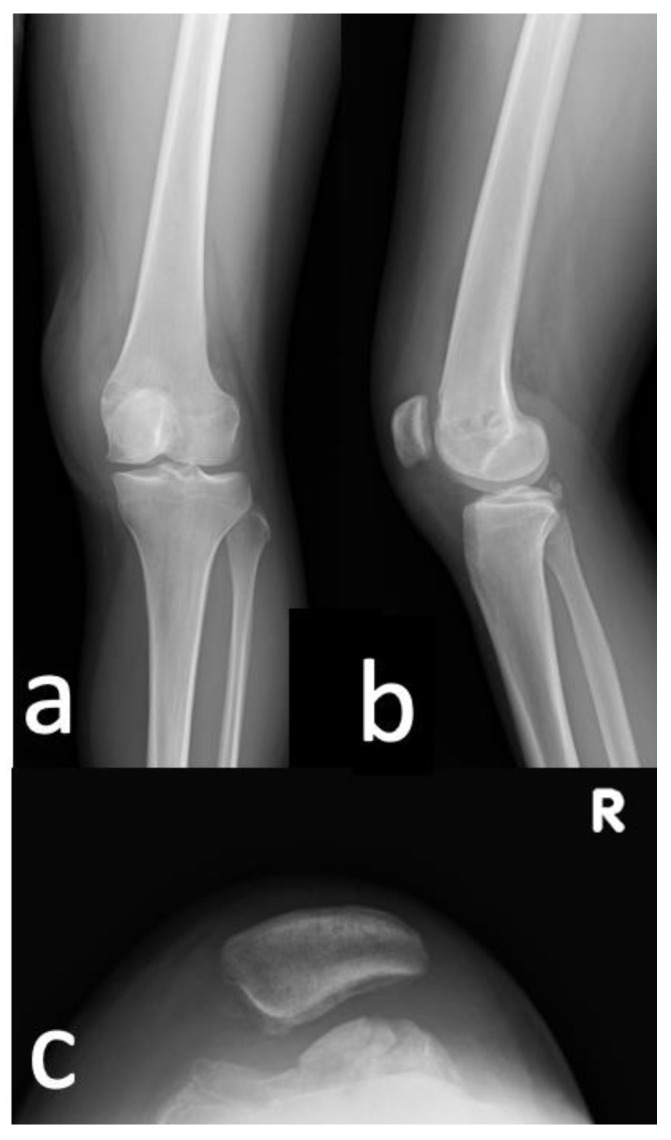
(**a**) Anteroposterior, (**b**) lateral and (**c**) merchant knee plain radiographs of the patient at postoperative 2 months.

**Figure 10 medicina-59-00986-f010:**
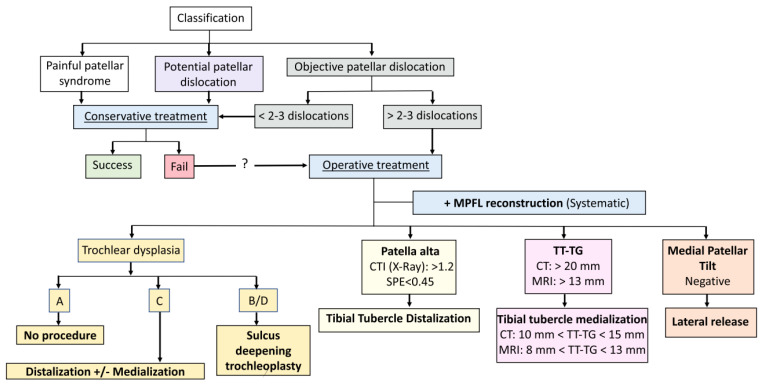
Dejour Treatment Guidelines “Menu a la carte”, 2021.

**Figure 11 medicina-59-00986-f011:**
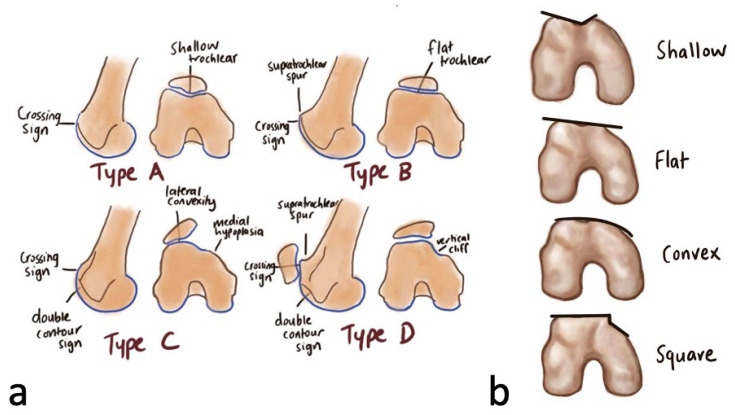
(**a**) Dejour classification of trochlea dysplasia. (**b**) The four shapes of the trochlea in correspondence to Dejour classification.

## Data Availability

Not applicable.

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
