# Peer review of "Reconstruction of High-Grade Trochlea Dysplasia in a Young Female with Recurrent Patella Dislocation: A Case Report"

_medicina, 2023, doi:10.3390/medicina59050986_

Round 1

Reviewer 1 Report

I've read this article titled “Reconstruction of high-grade trochlear dysplasia in a young female with recurrent patellar dislocation: A Case Report. The authors presented one typical case of patellofemoral instability, the diagnostic process, the preoperative surgical planning and the surgical procedure. What I am missing in this article is the originality of the case. I am missing more information about the case, and why it is required to publish its course.

I think that the presented data are not adequate for publication as a “Case report” in a journal with such a high IF. It would be more suitable as a Letter to the editor or Communication.

Author Response

Dear Reviewer,

Thank you for your review. We appreciate your feedback. Whilst we agree that case reports should have originality, we also believe that case reports also serve educational purposes. The aim of this case report is to provide an overview of patellofemoral instability for the readers and highlight the use of an easy-to-follow treatment algorithm due to the complexity behind the anatomy and biomechanisms of patellar instability. By using a typical case allows readers to be able to generalize and incorporate the findings from this case report towards their own patients. We have also highlighted controversial issues such as the reliability of the sulcus angle as an accurate parameter for diagnosis of trochlear dysplasia, which has not been previously mentioned in the literature.

Reviewer 2 Report

Dear authors it was a great time while going through your case report. It was done very well. Few comments and suggestions are there.

Dear authors:
It is my pleasure to review your excellent case report, abstract is presented very well. Methodology also explained very well. I have few comments to improve the case report.

1.    Abstract – Is ok
2.    Methodology- What was the follow-up period of this case. If you have, please write in the methodology. When was the study done? Write it in Clinical case section.
3.    Results are explain well in detailed.
4.    Please write down the limitations of your case report.
5.    Can you generalized you results on other subjects having same diagnosis?
6.    Do you recommend MQTFL reconstruction for the subjects having recurrent patellar dislocation, if so please write this in your conclusion and abstract.
7.    Abbreviations table can be added or next to every new abbreviations write it full name in the text.
8.    References- Year of the publication should me made Bold.
9.    Institutional Review Board Statement: The study was conducted in accordance with the Declaration of Helsinki and approved by the Institutional Review Board of Chang Gung Medical Foundation Institutional Review Board, IRB 202002554A3C501.- please write date of ethical approval.

Best regards

Author Response

  1.    Abstract – Is ok
  2.  Methodology- What was the follow-up period of this case. If you have, please write in the methodology. When was the study done? Write it in Clinical case section.

Reply: The follow-up period was 6 months to date. We have revised this. Thanks for the comment.

3. Results are explain well in detailed.

Reply: Thanks for the comment.

  1. Please write down the limitations of your case report.

Reply: Thanks for the comment. The main limitation for this case report is the inclusion of one single case that is typical for patellofemoral instability. However, a typical case allows readers to generalise the results when they are presented with patients of similar clinical conditions. A follow up period longer than 6 months would be more satisfactory. It has been revised and included in the Discussions section.

  1. Can you generalized you results on other subjects having same diagnosis?

Reply: Thanks for the comment. We believe by including a typical case of patellofemoral instability, this allows readers to be able to generalize the results on their patients having the same diagnosis.

  1. Do you recommend MQTFL reconstruction for the subjects having recurrent patellar dislocation, if so please write this in your conclusion and abstract.

Reply: Thanks for the comment. We recommend MQTFL reconstruction as an alternative to MPFL reconstruction due to satisfactory clinical and patient reported outcomes and a reduced risk of iatrogenic patellar fracture. The statement has been revised in conclusion and abstract.

  1. Abbreviations table can be added or next to every new abbreviations write it full name in the text.

Reply: Thanks for the comment. This has been corrected in the revised manuscript.

  1. References- Year of the publication should me made Bold.

Reply: Thanks for the comment. This has been amended in the revised manuscript.

  1. Institutional Review Board Statement: The study was conducted in accordance with the Declaration of Helsinki and approved by the Institutional Review Board of Chang Gung Medical Foundation Institutional Review Board, IRB 202002554A3C501.- please write date of ethical approval.

Reply: Thanks for the comment. The date of ethical approval was March 14th, 2022, which has been included in the revised manuscript.

Round 2

Reviewer 1 Report

After some significantly minor revisions provided by the authors - I am still convinced, that article is missing any signs of originality and that all presented data are not adequate for publication as a “Case report” in a journal with such a high IF. It would be more suitable as a Letter to the editor or Communication.

Author Response

Thanks for the comment. We think the paper is more concise after the two reviewers' and editors' valuable opinions. We hope the case report can provide some educational purposes and also originality.